# Exploring the Use of Viral Vectors Pseudotyped with Viral Glycoproteins as Tools to Study Antibody-Mediated Neutralizing Activity

**DOI:** 10.3390/microorganisms13081785

**Published:** 2025-07-31

**Authors:** Miguel Ramos-Cela, Vittoria Forconi, Roberta Antonelli, Alessandro Manenti, Emanuele Montomoli

**Affiliations:** 1Vismederi Research s.r.l., 53100 Siena, Italy; 2Department of Molecular and Developmental Medicine, University of Siena, 53100 Siena, Italy

**Keywords:** lentivirus, vaccines, monoclonal antibodies, pseudotyped virus, neutralization assays

## Abstract

Recent outbreaks of highly pathogenic human RNA viruses from probable zoonotic origin have highlighted the relevance of epidemic preparedness as a society. However, research in vaccinology and virology, as well as epidemiologic surveillance, is often constrained by the biological risk that live virus experimentation entails. These also involve expensive costs, time-consuming procedures, and advanced personnel expertise, hampering market access for many drugs. Most of these drawbacks can be circumvented with the use of pseudotyped viruses, which are surrogate, non-pathogenic recombinant viral particles bearing the surface envelope protein of a virus of interest. Pseudotyped viruses significantly expand the research potential in virology, enabling the study of non-culturable or highly infectious pathogens in a safer environment. Most are derived from lentiviral vectors, which confer a series of advantages due to their superior efficiency. During the past decade, many studies employing pseudotyped viruses have evaluated the efficacy of vaccines or monoclonal antibodies for relevant pathogens such as *HIV-1*, *Ebolavirus*, *Influenza* virus, or *SARS-CoV-2*. In this review, we aim to provide an overview of the applications of pseudotyped viruses when evaluating the neutralization capacity of exposed individuals, or candidate vaccines and antivirals in both preclinical models and clinical trials, to further help develop effective countermeasures against emerging neutralization-escape phenotypes.

## 1. Introduction

Humanity has always endured infectious diseases, but in recent years, viral epidemics have surged at an unprecedented rate, largely driven by increasing global connectivity, man-made environmental disruptions, and intensified interactions with animal reservoirs. Deforestation, climate change, and urban expansions have facilitated cross-species transmission, enabling viruses to spread into new populations and geographic regions. Ribonucleic acid (RNA) viruses in particular pose a significant risk due to their higher mutation rates, which allow them to adapt efficiently to new hosts and be maintained subclinically in reservoirs. Historical examples, such as the role of pig–duck farming in *Influenza* virus outbreaks, the spillover of *Human Immunodeficiency Virus* (*HIV*) from primates to humans, or the likely animal origin of the *Severe Acute Respiratory Syndrome Coronavirus 2* (*SARS-CoV-2*), illustrate how zoonotic viruses can trigger widespread epidemics [1].

Despite our vast and increased knowledge on molecular virology, epidemiologic trends, and availability of preventive strategies, the emergence and re-emergence of viral pathogens in the 21st century strongly suggests that the danger of having new pandemics may persist over time. The recent *SARS-CoV-2* outbreak underscored the urgent need for improved surveillance, rapid diagnostic tools, and global coordination in epidemic preparedness. Strengthening laboratory networks, investing in sequencing technologies, and enhancing antigenic characterization capabilities will be critical for future outbreak response. Beyond scientific advancements, governments must recognize epidemic preparedness as a strategic priority, requiring sustained funding and policy commitment to mitigate emerging biological threats [2].

In this sense, serological assays play a vital role in monitoring viral transmission patterns and assessing the effectiveness of large-scale immunization efforts by detecting antibodies targeting specific viral surface proteins. They also stand at the center of preclinical and clinical drug development, since they are required by regulatory institutions to assess early product efficacy during clinical trials and in real-world evidence studies [3]. Regarding viruses that undergo a viremic phase, vaccine-induced immunity primarily protects through the generation of antibodies, as these pathogens reach their target tissues by crossing through the bloodstream in an extracellular state. Although this tends to be determined by quantification of serum antibodies, there may be many adaptive immune responses involved that potentially correlate with protection (mucosal antibodies, T-cell immunity, etc.). In most cases, antibodies serve as correlates of protection against infection, whereas T cell immunity is a correlate of protection against disease. However, not all antibodies that bind the antigen are the same. Antibody-binding assays are generally derived from Enzyme-Linked Immunosorbent Assay (ELISA) tests for convenience and costs, but these do not necessarily have neutralization functions and prevent infection. Also, the subclass of Immunoglobulin G (IgG) plays an important role in protection [4]. All this has led to the use of more complex techniques, such as neutralization assays, to evaluate the efficacy of vaccines against viral diseases during their clinical and preclinical development. These assays assess not only the binding of the antibody to the virus but also its ability to prevent infection and are frequently required by pharmaceutical companies and regulatory agencies to advance clinical development.

Nonetheless, traditional serological tests have several limitations, being restrained by operational costs, required expertise, biosafety concerns, and interlaboratory variability. Live-virus neutralization assays for pathogens such as Ebola virus (*EBOV*) require highly inaccessible level 4 biocontainment facilities, which greatly limit the capacity of rapid serological diagnosis in endemic and/or low-income settings. When dealing with emerging viruses, isolating specific variants to study the phenotypic effect of mutations can be extremely challenging (if not impossible). It is in this context that novel techniques such as pseudotyped viruses have enabled their use as alternative agents for conducting neutralization assays, simplifying the study of highly pathogenic viruses and providing accessibility to a greater number of research centers [3]. Two of their most interesting features are their capacity to be safely handled in Biosafety Level 2 (BSL-2) laboratories and the fact that they often carry a marker gene (such as *luciferase* or *Green Fluorescent Protein*), which allows for accurate quantification [5]. This has greatly shortened and improved the process of serological surveillance in epidemic contexts and vaccine development trials, ultimately contributing to our unified efforts to combat human infectious diseases.

## 2. Overview and Definition of Pseudotyped Viruses

Pseudotyped viruses are chimeric viruses composed of a viral core with a lipid envelope bearing the surface protein(s) of a heterologous virus [6]. These are widely employed as vectors for gene delivery strategies into targeted cells or tissues in vaccination [7] or gene therapy [8,9], among many other uses. However, typically when referring to pseudotyped viruses, the focus is set on viruses that contain a surface protein of interest of a target virus, expressed on a parent virus with an incomplete recombinant genome. The surface protein is normally the one that mediates binding and entry into host cells by attaching to the cell surface and inducing fusion of the viral envelope with the plasma membrane [10]. Therefore, a pseudotyped virus is generally a chimeric viral particle whose surface protein (the envelope protein of enveloped viruses or the capsid protein of non-enveloped viruses) differs from its core component. This targeted viral protein has a similar structure to that of the native live virus and, most importantly, can mediate viral entry into susceptible cells. Since the pseudotyped virus harbors a modified genome with transgenes, it will normally lead to a defective replication cycle in the infected cells. For all these reasons, pseudotyped viruses confer a series of superior advantages compared with native viruses. Firstly, it can be easily constructed based on the target protein sequence. If the gene sequence is publicly available, it can be synthesized and cloned into an adequate expression plasmid and then expressed within the proper viral vector, avoiding the use of the real virus. Secondly, since the genome lacks (some) essential viral replication genes, they do not generate major biosafety concerns and can be manipulated in level 2 biosafety laboratories, which circumvents the need for restrictive, expensive, and widely unavailable higher-category biosafety facilities, also requiring highly trained professionals. Thirdly, they can be quantified precisely, because part of the genetic sequence of the viral vector (generally coding the structural glycoprotein) is replaced with a reporter gene, which provides a measurable signal right after cell infection. Lastly, as the foreign protein retains the tridimensional structure and biological function of the authentic virus, the pseudotyped virus can be used as a reliable alternative when studying viral entry and binding properties of antibodies targeting the surface protein of interest [11,12].

## 3. Vectors Used for Pseudotyping

### 3.1. Lentiviral Vectors

As of today, lentiviral vectors stand at the forefront of pseudotyped virus research and applications, mainly due to the capacity of transducing both dividing and non-dividing cells, as well as their extra genetic elements that confer a higher transduction efficiency [12]. These vectors confer a series of advantages over other systems, such as transgene integration in the host genome that provides a heritable and sustained expression over time, their ability to package relatively large sequences, low immunogenicity, acceptable biosafety, and large plasticity to be pseudotyped with numerous heterologous glycoproteins [8,13,14]. Almost all lentiviral vectors are derived from *HIV* type 1 (*HIV-1*).

*HIV* is an enveloped spherical retrovirus, about 80 to 100 nm in diameter, which encloses two molecules of genomic single-stranded RNA and the required enzymes for completing a full replicative cycle: the reverse transcriptase, the integrase, and the protease (see Figure 1). The lipid bilayer that surrounds the capsid is derived from the host cell membrane and has exposed surface glycoproteins, gp120, anchored to the virus via the transmembrane protein, gp41, among other host proteins. The non-covalent union of gp120 and gp41 forms an unstable heterodimer, which associates with two identical heterodimers to form a trimer that constitutes the viral spike [15]. This spike interacts through its gp120 unit with the lymphocytic receptor CD4, which upon a viral protein rearrangement and a second coreceptor contact (CCR5 or CXCR4), triggers membrane fusion and release of the viral core into the cytoplasm [15].

The first use of *HIV* as a genetic engineering tool was within gene delivery strategies, using a replicative vector that had some of its genes replaced with transgenes, aimed at achieving a successful gene therapy. Nevertheless, due to the pathogenic potential that a replication-competent *HIV* vector entailed, several safety improvements have been made since then. Lentiviral vectors are generated by co-transfecting eukaryotic cells with a plasmid coding for a truncated form of the genomic RNA and additional plasmid coding for structural proteins. The *HIV* genome is partially cloned into a packaging plasmid (containing structural and essential accessory genes), a genomic plasmid (containing the packaging signal and the reporter gene), and an envelope plasmid (carrying the non-*HIV*, envelope protein coding sequence). The structure of the plasmids and which viral proteins are provided vary depending on which generation of lentiviral vector is chosen. At least three different generations of lentiviral vectors have been designed and employed globally, which are summarized in Figure 2. Some variations in these strategies do exist, such as simpler two-plasmid systems or *Simian Immunodeficiency Virus* (*SIV*)-derived vectors to provide further biosecurity [12].

Each of the three generations of *HIV-1*-derived lentiviral vectors aimed to address specific biosafety and functional concerns inherent to working with a human retrovirus. The first-generation packaging systems maintained most of the *HIV-1* genome across only two or three plasmids and included several accessory genes such as *vif*, *vpr*, *vpu*, and *nef*, raising significant biosafety concerns due to the potential for recombination and generation of replication-competent lentiviruses [13]. For second-generation vectors, these non-essential accessory genes were removed, and the *tat* gene was retained, with transcription still relying on the *HIV-1 LTR* promoter. This version improved biosafety while maintaining vector efficiency. The major leap in safety and modularity came with third-generation vectors, which introduced several innovations: *tat* dependence was eliminated by replacing the *HIV-1 LTR* promoter with a constitutively active promoter (such as *Cytomegalovirus* (*CMV*) or *Rous Sarcoma Virus*), the *rev* gene was segregated onto a separate plasmid, and the structural and enzymatic genes (*gag*, *pol*) were further separated from the envelope gene, resulting in a four-plasmid system. Additionally, self-inactivating (SIN) *LTRs* were introduced, preventing reactivation of the integrated provirus and reducing the risk of insertional mutagenesis [13].

Further safety and performance features were incorporated over time into third-generation systems, such as the *Rev Response Element* (*RRE*) for efficient nuclear export, the *central polypurine tract* (*cPPT*) for improved nuclear import, and the *Woodchuck Post-transcriptional Regulatory Element* (*WPRE*) to enhance transgene expression. These advancements not only greatly minimized the possibility of replication-competent lentiviruses formation but also improved transduction efficiency in non-dividing cells (a current hallmark of lentiviral systems). As a result, third-generation lentiviruses are now considered the safest and most versatile standard for a wide range of molecular biology, immunology, and cell engineering applications, offering a strong balance between high-level expression and regulatory compliance [14].

Regarding the use of lentiviral vectors for pseudotype generation, the formation of pseudotype particles after cell transfection mimics a natural *HIV* cycle, with viral *U3* promoter guiding the expression of viral proteins and the *Psi* (Ψ) packaging signal driving the encapsidation of genomic RNA into nascent particles. In infected cells, the *RRE* guides the nuclear export of viral mRNA, leading to the expression of the reporter gene. Other first-generation vector systems (e.g., *pCMV* precursor), alternatively, possess functional *vif*, *vpr*, *vpu*, and *nef* genes [16,17]. Second-generation lentiviral *HIV* (*p8.91* or *pCSFLW* variants) makes use of modified and truncated forms of *HIV-1* proviruses for additional biosafety purposes, lacking all of the *Env*, *vif*, *nef*, *vpu*, and *vpr* genes and the Ψ signal. Some also have safety deletions in the *U3* promoter, creating the so-called self-inactivating vectors. The remarkable feature of third-generation vectors is that the packaging plasmid is split into two, providing the *rev* gene separately, and frequently, the 5′ *U3* sequence of the genomic plasmid is replaced with other promoters. Fourth-generation vectors further divide coding sequences for enhanced biosafety but vary significantly from one another [16,17,18]. A summarized diagram of pseudotype generation using a typical three-plasmid system is shown in Figure 3.

*HIV* virions acquire their lipid envelope from the host cell’s plasma membrane, budding at specific regions named lipid rafts enriched in cholesterol and immune receptors. This strategy minimizes immune detection and degradation. The affinity of transmembrane proteins for lipid rafts influences their presence in lentiviral particles, making post-translational modifications, folding, and intracellular pathways crucial factors when pseudotyping lentiviral vectors. Acylation, for instance, enhances protein recruitment to lipid rafts, improving pseudotyping efficiency [16,19].

The highly immunogenic nature of *HIV*, its insertional preference among cellular genes and potential disruption of host gene expression, and its significant biosafety concerns, when used for clinical purposes, prompted the development of integrase-defective lentiviral vectors unrelated to *HIV*. This led to *SIV*-derived vectors, a nonhuman lentivirus that is harmless both in its natural host and in humans. *SIV*-based vector pseudotypes use last-generation four-plasmid systems, which may be more suitable for vaccine gene delivery strategies [20].

### 3.2. Gammaretrovirus Vectors

Other vectors from the *Retroviridae* family, such as the *Murine Leukemia Virus* (*MLV*) or *Feline Immunodeficiency Virus* (*FIV*), have also been employed as viral vector systems in both gene therapy and serological assays, sharing extensive homology with the *HIV-1* platform [21,22]. Despite these similarities, some envelope proteins pseudotype worse into *HIV* vectors, due to subtle differences in intracellular viral assembly sites as well as interactions between *Env* and core proteins, which determine the recruitment of the glycoprotein into the nascent pseudoviral particle [23]. *MLV* is a genetically simpler retrovirus that does not encode regulatory or accessory genes and achieves high transduction efficiency and RNA expression levels. Nevertheless, it is only capable of infecting dividing cells, and therefore, its employment is restricted to highly proliferative cell lines. Besides its extensive use in neutralization assays, *MLV*-derived vectors have been quite successful in gene therapy approaches, and many are being tested in clinical trials for monogenic diseases. The main concern is the insertional preference of the transgene close to the transcriptional start site, frequently involving upregulation of neighboring genes and also gene disruption [24].

### 3.3. The VSV Vector

The *Vesicular Stomatitis Virus* (*VSV*) is a widespread animal pathogen that causes livestock diseases endemically in Central and South America, with occasional outbreaks in the US. *VSV* contains a single-stranded, negative-sense 11 kb RNA genome very suitable for genetic manipulation, since it exhibits broad cell tropism, rapid replication, and renders high virus yields and levels of transgene expression [25]. Biotechnological applications of *VSV* generally rely on reverse genetics, which manipulates the *VSV* genome from artificial complementary DNA (cDNA) plasmids, and when transcribed in the host cell, fully infectious *VSV* particles can be recovered. The *VSV* system is advantageous in that it provides increased flexibility for the pseudotyped envelope protein, allowing a greater range of parent viruses and further possibilities of targeted tropism. Because of the intrinsic characteristics of the vector, there may be residual infectious *VSV* virions mixed with the final pseudotyped virus pool, possibly complicating the evaluation of the neutralization assay [11,26].

As in most other pseudotyped viruses, *VSV* recombinant vectors can be replication incompetent or replication competent. In replication incompetent vectors, the *VSV* glycoprotein gene is deleted and provided in trans with a non-*VSV* heterologous glycoprotein gene, rendering defective particles only capable of replicating for a single round unless the glycoprotein is supplied continuously. On the other hand, in replication-competent vectors, the glycoprotein gene is either kept or replaced with a heterologous sequence but always within the genome, producing propagative vectors useful for analyzing viral entry pathways, screening mutants prone to immune evasion, and developing replicative viral vaccines for continuous supply of the desired antigen [11,25].

### 3.4. Other Packaging Systems

Sometimes, the aforementioned vectors have been proven inefficient when pseudotyping the virus with specific envelope proteins from relevant human pathogens. While several other packaging systems have been engineered, and even more are currently under development, only some of the most commonly employed in biomedical research have been outlined (see Table 1). In the last 10–20 years, *Foamy Virus* (*FV*) or *spumaretrovirus* vectors have emerged as a reliable alternative to other retroviruses. Regarding their natural host, they are not endemic in humans but are prevalent and non-pathogenic in nonhuman primates and other mammals. Their preliminary success is owed to their broad tropism, large transgene capacity, improved integration profile, and ability to mediate stable gene expression, making them valuable for therapeutic applications. They require mitosis for efficient transduction, limiting their use in non-dividing cells, but remain ideal for ex vivo applications like hematopoietic stem cell gene therapy. *FV* vectors have built-in safety features, can be produced at high titers without contaminating replication-competent viruses, and are easy to establish in most laboratories. They have a favorable safety profile compared to *MLV* and *HIV-1* vectors, with no reported adverse events in preclinical models. Given their efficacy and safety data, *FV* vectors are strong candidates for clinical trials targeting severe hematopoietic or immunodeficiency disorders, where their therapeutic benefits may outweigh potential risks [27,28,29].

In some cases, self-assembly pseudoviral systems for enveloped viruses have been designed, where structural proteins (in addition to the envelope protein) are expressed and co-assembled with manipulated plasmids carrying reporter genes. This approach has the advantage that it more closely resembles the native structural environment of the virus, which may facilitate the study of the infective cycle or the phenotypic effect of acquired mutations. However, it may not be suitable for all enveloped viruses and has only been developed for certain examples. Self-assembled *Nipah* viruses can be formed by heterologous expression of matrix and envelope proteins [30]. Similarly, S, M, and N proteins are enough for empty *SARS-CoV-2* virus-like particle assembly [31]. Other successful examples of self-assembly pseudoviruses include the *Dengue* virus and the *West Nile* virus, which have been harnessed for drug-discovery screening and antibody-mediated neutralizing activity, respectively [32,33]. In these flavivirus cases, generation of virus-like particles (VLPs) is achieved by transfection of a genomic RNA plasmid with a reporter gene, with the structural genes provided in trans in a second plasmid. *Coxsackievirus* pseudotypes for many different pathogenic strains have also been developed using reverse genetics strategies, rendering replication-defective viruses suitable for neutralization antibody assays [34,35,36].

### 3.5. Non-Enveloped Pseudotyped Viruses

Non-enveloped viruses feature a less intricate structural arrangement, which also simplifies the rationale of viral pseudotyping. Regardless of their structural complexity, all viruses contain at least one protein layer, the capsid, which encases the nucleic acid. This capsid is the most external layer in non-enveloped viruses and lacks both host plasmatic membrane lipids and additional embedded glycoproteins. In many cases, such as *Human papillomavirus* (*HPV*) or *Hepatitis B virus* (*HBV*), heterologous expression of only one structural protein is enough for self-assembly of empty VLPs, this being the case for many vaccine design approaches [37]. Envelope-free pseudotyped viruses are engineered based on a simple two-expression plasmid system: one eukaryotic expression plasmid that codes the structural viral protein, and a viral genome plasmid where the structural protein Open Reading Frame (ORF) is replaced by a reporter gene. When eukaryotic cells are co-transfected with both plasmids, naked pseudotyped viruses antigenically and structurally similar to the wild-type virus are formed, while harboring a defective genome with a reporter gene [12].

**Table 1 microorganisms-13-01785-t001:** Some examples of pseudotyping systems for relevant human viral pathogens and their uses.

Packaging System	Viral Protein	Research Line	Study
*HIV-1*	*EBOV* GP	Gene therapy, cellular tropism, antiviral screening, cross-neutralization activity	[38,39,40]
	*HIV Env*	Cellular tropism, neutralization antibody assay, Env mutation screening, antiretroviral therapy resistance, receptor recognition	[41,42,43,44]
	*CHIKV* E3 E2 E1	Neutralization antibody assay, cellular tropism, in vivo imaging model	[45,46]
	*RSV* SH/G/F	Neutralization antibody assay, antiviral screening, identification of neutralization-escape genotypes.	[47]
	*SARS-CoV-2* Spike	Receptor recognition and virus tropism, neutralization antibody epitopes, immune evasion, viral evolution	[48,49,50,51,52,53,54]
	*MERS-CoV* Spike	Receptor recognition, neutralization antibody assay, screening inhibitor of viral entry, viral evolution	[55,56]
	*Influenza* virus HA, NA, M2	Mechanism of virus entry, antiviral screening, neutralization antibody assay	[17]
	*Lassa* virus	Neutralization antibody assay	[57]
	*DENV* PrM/E	Mechanism of virus entry, antiviral screening	[58]
	*Zika* virus PrM/E	Antiviral screening, oncologic virotherapy	[59,60]
	*RVFV* GP	Neutralization antibody assay	[61]
	*Rabies* virus G	Neutralization antibody assay, serosurveillance, cross-neutralizing activity, drug screening, gene delivery	[62,63,64,65,66,67]
	*HBV* GP	Entry inhibitor development	[68]
*MLV*	*Influenza* HA	Neutralization antibody assay	[17,69]
	*HIV Env*	Antiviral and antibody screening	[70]
	*EBOV* GP	Mutation study, infection mechanism, human spillover	[71,72,73,74]
	*MERS-CoV* Spike	Virus entry	[22,75]
	*SIV Env*	In vivo gene therapy for AIDS	[76]
	*HCV* GP	Infection mechanism, neutralization antibody assay, epitope mapping	[77,78,79]
*VSV*	*EBOV* GP	Vaccine development, viral tropism, neutralization antibody assay, GP function	[80,81,82,83]
	*NiV Env*	Antiviral screening, mutation study, neutralization antibody assay	[84,85,86]
	*CCHFV* GP	Neutralization antibody assay	[87]
	*CHIKV* E1 E2	Neutralization antibody assay	[88]
	*Measles* virus H, F	Vaccine development	[89]
	*Avian Influenza* HA, NA	Antiviral screening, neutralization antibody assay	[90]
	*HIV*	HIV vaccine development	[91]
	*SFTS* virus GP	Neutralization antibody assay, vaccine development, infection mechanism	[92,93,94]
	*HBV* GP	Entry inhibitor development	[68]
*PFV*	*HIV* specific epitopes	HIV vaccine development, AIDS gene therapy	[29]
*HPV*	*HPV* L1/L2	Neutralization antibody assay, standardized vaccine efficacy	[95,96]
Self-assembly/ reverse genetics	*PV* capsid	Neutralization antibody assay, antigenic analysis	[97]
	*NiV* M/*Env*	Structural analysis	[30]
	*DENV* C, prM, E	Drug discovery	[32]
	*SARS-CoV-2* S, M, N	Viral evolution, infection mechanism	[31]
	*WNV*	Neutralization antibody assay	[33]
	*Coxsackievirus*	Neutralization antibody assay	[34,35,36]

*CCHFV*: Crimean–Congo hemorrhagic fever virus; *CHIKV*: Chikungunya virus; *CoV*: coronavirus; *DENV*: Dengue virus; *EBOV*: *Zaire Ebolavirus*; F: fusion protein; G: glycoprotein; GP: glycoprotein; H: hemagglutinin; HA: hemagglutinin; *HBV*: hepatitis B virus; *HCV*: hepatitis C virus; *HIV*: human immunodeficiency virus; *HPV*: human papillomavirus; MERS: middle-east respiratory syndrome; NA: neuraminidase; *NiV*: Nipah virus; *PFV*: prototype foamy virus; *PV*: polio virus; *RSV*: respiratory syncytial virus; *RVFV*: Rift Valley fever virus; SARS: severe acute respiratory syndrome; *SFTS*: severe fever with thrombocytopenia; SH: small hydrophobic protein; *SIV*: simian immunodeficiency virus; *WNV*: West Nile virus.

## 4. Applications of Pseudotyped Viruses

The infection process of most viruses hinges on the physical, non-covalent interaction between a capsid or envelope protein from the virus and a specific cell surface receptor, which upon contact, ends up inducing the liberation of the viral nucleic acid into the cell (through many possible mechanisms). Pseudotyped viruses aim to mimic the infection process of wild-type viruses. Virus tropism is determined by the pseudotyped envelope protein, serving as a surrogate infective particle. Neutralizing antibodies typically target the viral envelope protein, which is responsible for interacting with the cellular receptor and facilitating viral entry and blocking this interaction. Therefore, pseudotyped viruses are frequently employed to evaluate the neutralizing activity of antibodies, regardless of their origin, and perform an in-depth analysis of the initial steps of virus infection. Moreover, the flexibility of pseudotyped viruses to be genetically engineered and their suitability for manipulation within BSL2 facilities circumvent the impossibility of culturing some live viruses or the complex acquisition of emerging variants (see Figure 3).

The reliability and applicability of downstream assays utilizing pseudotyped viruses stem from the strong correlation and replaceability with traditional live virus methods. With few exceptions, pseudotype-based neutralization assays have been proven as reliable alternatives to study antibody-mediated neutralizing activity or screening other entry inhibitors, as their neutralization curves tend to be equivalent to live virus assays [98]. While conventional live virus research techniques are unlikely to ever be substituted, assays based on pseudotyped viruses have quickly emerged as useful tools to elude most of the drawbacks that live virus research involves, especially after the *SARS-CoV-2* pandemic. Over the past few years and following an increased trend globally, many studies have made use of pseudotyped viruses to explore mechanisms of viral pathogenesis and infection, evaluate the efficacy of vaccines and monoclonal antibodies (mAb), screen antiviral drugs, analyze the antigenicity properties and glycosylation patterns of certain viral variants, and predict antibody-dependent cell-mediated cytotoxicity, among others. Below, the main research lines, in which pseudotyped viruses have been successfully employed, have been outlined.

### 4.1. Study of Viral Tropism and Cell Infection

Receptor recognition and subsequent interactions with host cell proteins constitute the primary steps of a viral replication cycle and a necessary condition for viral entry, which in most cases, is mediated by the envelope protein. These steps also determine relevant regulatory roles regarding viral host range, tissue tropism, and viral pathogenesis. Pseudotyped viruses have been frequently employed to understand the process of viral infection and genome entry into the host cell. A recent illustrative example of worldwide relevance was the evidence that host cell factors Angiotensin-converting enzyme 2 (ACE2) and Transmembrane Serine Protease 2 (TMPRSS2) play crucial roles in the early stages of *SARS-CoV-2* target cell entry, discovered by means of *VSV* particles pseudotyped with coronavirus S protein [53]. Regarding the S protein proteolytic cleavage, pseudotyped viruses were used to describe how the *Omicron BA.1* variant has altered cellular entry mechanisms, relying less on the TMPRSS2-mediated plasma membrane fusion pathway and more on the endocytic pathway, which is linked to its reduced replication in lung and gut cells. These findings help explain *Omicron*’s altered infectivity, fusogenicity, and potentially lower severity compared to *Delta* [51].

Cell tropism of many other viral pathogens, especially emerging viruses, has been elucidated with the use of pseudotyped viruses, highlighting the relevance of superior accessibility to study the infection process. In the case of *EBOV*, it helped researchers understand how the soluble form of its envelope glycoprotein impairs neutrophil activation, while the transmembrane form displays selectivity for endothelial cells to trigger the hemorrhagic symptoms of the disease [74,83]. Another study described the shared amino acid regions from the receptor binding domain of filovirus GP, suggesting pan-neutralizing filovirus entry inhibitors or vaccines may be feasible [72]. For the *Chikungunya* virus, its broad-spectrum tropism in different human cell populations, such as the blood-brain barrier, lung cells, and joint tissues, was demonstrated and even extended to other species of mammals. Given the rapid spread of the Reunion Island outbreaks, this provided early evidence that an airborne transmission mechanism was possible [45]. *HIV*-vectored *Dengue* pseudotypes were employed to understand the cleavage of PrM protein during entry and its molecular characterization between different serotypes, potentially affecting their infectivity [58].

### 4.2. Assessment of Antibody-Mediated Neutralizing Activity

The gold standard application of pseudotyped viruses may be to evaluate the neutralizing activity of antibodies from serum samples or immunotherapies, with the pseudovirion (or pseudotype)-based neutralization assay (PBNA) (see Figure 4). Although there may be immunological mechanisms involved in viral clearance and preventing disease onset, for most viruses, surpassing a certain threshold of specific neutralizing antibody levels generally serves as a correlate of protection (indicator of immune protection), mostly against infection rather than disease [4,52]. Vaccine-induced immunity is primarily mediated by the generation of neutralizing antibodies by plasmatic B lymphocytes and provides protection through this mechanism. For enveloped viruses, neutralizing antibodies are primarily targeted toward the envelope protein, preventing infection by blocking receptor binding and/or viral fusion. Therefore, these neutralizing antibody levels can be employed to analyze either vaccine efficacy or the immune protection of individuals who have been exposed to the viral antigen. Classical immunological antibody-binding assays (ELISA, immunofluorescence, Western Blotting, microarrays, etc.) enable the detection of surface glycoprotein directed antibodies present in sera or other samples (nasal, vaginal, etc.), but they cannot discriminate whether these antibodies can neutralize virus infection, since they evaluate only binding ability and not antibody function [4]. Hence, there is a medical need to determine neutralizing antibody titers rather than only specific antibody presence, which may correlate worse with disease protection.

PBNA has been employed thousands of times to understand disease protection and vaccine evaluation against many different viruses and has largely replaced the classic plaque reduction neutralization test, which necessarily involved inconvenient BSL3/BSL4 facilities. Pseudotypes can be used not only to evaluate vaccines already available on the market but also for the early stages of vaccine development with raw candidates and, less frequently, to evaluate the neutralization capacity of diverse monoclonal antibodies. Viral diversity can be reproduced by generating pseudotype libraries, which can be useful to evaluate broad-spectrum vaccines or to predict the potential protective activity of neutralizing antibodies [40,41,99,100,101]. Moreover, they have been extensively employed for serosurveillance, aiming to determine the prevalence of neutralizing antibodies in regions affected by epidemic outbreaks and gaining epidemiologic insights to shape vaccination policies [64,102,103,104].

During the initial phases of the *SARS-CoV-2* pandemic, early pseudotype-based neutralization assays were used to analyze large panels of human monoclonal antibodies targeting the spike protein, among which several showed protective activity as immunotherapy in animal models and cross-neutralization from other coronavirus species [48,105]. Initially, novel emerging *SARS-CoV-2* variants with mutations in immunodominant domains of the spike protein were described as neutralization escape phenotypes from several classes of therapeutically relevant antibodies and convalescent sera [106]. As the first vaccines came to market and universal efforts were made to comprehend correlates of protection, neutralizing antibody levels from vaccinees and convalescent cohorts were shown to be highly predictive of immune protection [52]. However, total loss of function against emerging *Omicron* sublineages (*XBB.1.5.70*) has now been reported [107]. Some groups even succeeded in creating a library of lentivirus-based pseudotypes using the spike proteins of all human coronaviruses, attempting to unravel the extent of pre-existing immunity induced by seropositivity to endemic seasonal coronaviruses and the impact of cross-reactivity on COVID-19 disease immunopathology [100]. Regarding direct *SARS-CoV-2* vaccine evaluation, neutralizing antibody responses have been measured very frequently using S-protein pseudotyped viruses for commercially available mRNA vaccines and those undergoing clinical development, especially to assess their efficacy against variants of concern [108,109,110,111]. The longitudinal evolution of type B lymphocytes in response to *SARS-CoV-2* variants has been analyzed with spike-pseudotyped lentiviruses, revealing differences between homologous and heterologous vaccination and showing how the antibody response to *Omicron*-based boosters is dominantly imprinted by the Wuhan original antigenic sin. This may provide relevant insights to inform policymakers on the use of heterologous vaccination strategies [112].

The advantage of using pseudotyped viruses for neutralization assays is maximized when handling extremely pathogenic viral families, which are difficult to isolate due to biosafety issues and would otherwise require BSL-4 settings. The Janssen adenovirus-based vaccine for *Zaire ebolavirus* (Ad26.ZEBOV) was prequalified in 2020 by the World Health Organization (WHO) and recommended by the WHO Strategic Advisory Group of Experts for prophylactic use, following positive results concerning the immunogenicity and protective efficacy determined by PBNA during phase 1 and 2 clinical trials [113]. *VSV*-vectored GP pseudotyped filoviruses were employed to discover a therapeutic cocktail comprising two broadly neutralizing human antibodies, which exhibited synergistic neutralizing activity and protection against *Zaire*, *Bundibugyo*, and *Sudan ebolavirus* strains [114]. Using pseudotyped *EBOV* GP lentivirus particles, isolated monoclonal antibodies from a human survivor of the 1995 Kikwit *Ebolavirus* disease outbreak also led to the discovery of a potent mAb with therapeutic potential against the 2014–2016 deadly *EBOV* strain [115].

Lentiviral *Env*-pseudotyped viruses have become the main endpoint neutralization assay used by National Institutes of Health (NIH-) sponsored *HIV* Vaccine Trials and others. For a recent *pox* vector *HIV* vaccine with subtype C inserts and a gp120 protein component, the magnitude and duration of vaccine-elicited immune responses from a prime–boost vaccine regimen designed against subtype C *HIV* were partially evaluated using this pseudotype neutralization assay [116,117]. A single dose of a potent monoclonal antibody targeting the *Env* CD4 binding site was evaluated with a lentiviral pseudotype library of 63 circulating *HIV* variants, showing suppression of viremia in *HIV-1*-infected individuals and evidence of enhanced host immunity [101]. Additionally, pseudotype-based neutralization assays have been employed to assess neutralizing activity against other emerging, pressing global health concerns, such as *Chikungunya*, *Yellow Fever*, *Japanese Encephalitis*, and *Lassa* virus, aiding in the complex task of developing much-needed effective and durable preventive strategies [57,118,119,120].

Most licensed Influenza vaccines target epitopes from the highly variable globular head of the hemagglutinin (HA) protein, aiming to block virus entry into the cells. However, this vaccine-elicited immune response confers primarily strain-specific protection and requires an annual process of vaccine development. In recent years, efforts to develop a universal next-generation Influenza vaccine have focused on eliciting cross-neutralizing antibodies targeting the stalk region of the HA protein, which is more antigenically conserved than its globular head. Traditional serological assays for Influenza have largely been replaced by pseudotyped virus-based assays due to their enhanced efficiency, sensitivity, and safety, facilitating the assessment of heterosubtypic protection against *Influenza* strains with pandemic potential [17,99,121]. Another interesting application of pseudotyped viruses is to evaluate immune responses to extinct diseases. A recent study using lentiviral pseudotypes from the eradicated *Influenza 1918 H1N1* pandemic strain suggested that older centenarian survivors of the 1918 Spanish Flu could be more resilient to COVID-19 and Influenza, experiencing asymptomatic or milder clinical manifestations [104].

### 4.3. Pseudotypes as Immunogens/Vaccine Vectors

Viral vectored vaccines have demonstrated great potential since they produce robust protective immunity by providing a continuous supply of the desired antigen. The first applications of viral vector vaccines date back more than 40 years with the *vaccinia* virus, which was used as a vector to express the Hepatitis B surface antigen of the hepatitis B virus [122]. Since then, many other viral families have been employed for vaccine development, including lentivirus, retrovirus, adenovirus, poxvirus, alphavirus, flavivirus, and others, most of which are still under early clinical evaluation. Several candidates hold the promise of delivering a new vaccine to the market for the prevention of diseases whose vaccine development programs have repeatedly failed so far, such as AIDS, hepatitis C, or pandemic Influenza [123].

For enveloped viruses, it may be worth mentioning replication-competent *VSV* pseudotypes as vaccine vectors for their ability to be pseudotyped, which has been proven extremely useful to express heterologous antigens in vivo. Most approaches insert additional envelope genes in the *VSV* genome and remove or attenuate the *VSV* glycoprotein gene, which is a key driver of neurovirulence. Some of the main advantages of *VSV* pseudotypes for viral vaccine development involve their low seroprevalence in humans (implying lack of pre-existing immunity), stability of transgene expression in target cells, lack of integration into the host genome, and suitability for genetic manipulation and pathogenic attenuation [124]. The *VSV*-vectored *Zaire Ebola* virus vaccine (rVSV∆G-ZEBOV-GP) was the first licensed viral vector vaccine for human use, following positive immunogenicity, efficacy, and safety results after pivotal phase III clinical trials [80]. Highly attenuated *VSV* vectors expressing *gag* protein as *HIV-1* vaccine candidates merited first-in-humans phase I clinical evaluation, rendering acceptable immunogenicity and safety profiles [125]. Moreover, two *SARS-CoV-2* spike pseudotyped *VSV* vaccine candidates engineered by Merck and the Israel Institute for Biological Research advanced early to clinical experimentation. The latter showed promising results in inducing neutralizing antibodies against all variants of concern and reached phase III trials, but ultimately, both clinical development programs were withdrawn [126,127].

### 4.4. Antiviral Drug Screening

One of the appealing applications of pseudotyped viruses lies in their suitability to be harnessed in high-throughput settings, making them valuable tools for drug discovery and to evaluate antiviral activity during the earliest stages of cell infection [128]. Besides monoclonal antibodies [48,67,101,105,114,115], multiple studies using pseudoviruses have sought to identify small-molecule compounds with therapeutic efficacy. In the context of emerging viral diseases, this approach involves screening existing approved medications for their potential to be repurposed in the event of an outbreak. Some notable examples include antiviral screening for *Nipah* virus [32], *Lassa* virus [129], *Dengue* virus [32], *Ebolavirus* and other filoviruses [130,131], and *Zika* virus [59], mostly based on *HIV*-lentivirus platforms.

Effective *HIV* entry inhibitors are scarce and of limited use, and both screening and the mechanisms involved in drug resistance have frequently been studied using pseudotypes. A large panel of primary *HIV-1 Envs* in lentivirus vectors to simulate genetic and antigenic diversities was developed to screen novel T20 derivatives with a higher therapeutic selectivity and genetic resistance barrier [132]. Fostemsavir was determined as a valuable therapeutic option for patients harboring multidrug-resistant *HIV* strains, by using pseudotyped viruses expressing the *Env* population from patient samples and assessing phenotypic susceptibility to antivirals [133]. The contribution of different gp120 V3 loop structure polymorphisms to maraviroc resistance was studied using a gp120 pseudotype *HIV* library [134]. While searching for additional entry inhibitors besides maraviroc and T20, Gossypol and its derivatives were tested using *Env*-pseudotyped *VSV* virions and demonstrated to inhibit early viral fusion, contradicting previous findings [135].

Regarding *SARS-CoV-2*, antiviral compounds targeting virus entry are lacking, and for those approved, there still exists a high risk of baseline resistance and unideal safety profiles, diminishing their clinical success against emerging variants [136]. Several aloperine derivatives were identified as promising anti-COVID-19 drug candidates by testing their anti-infective activity against *MLV*-spike pseudotyped virus. These drugs exhibited broad-spectrum inhibition among all variants of concern evaluated (including *D614G*, *Omicron*, and *Delta*), although *Omicron* subtypes were the least sensitive [137]. A high-throughput screening of 1800 licensed drugs using *SARS-CoV-2* pseudotyped lentivirus identified seven compounds suitable for COVID-19 clinical treatment, with clemastine showing the strongest anti-SARS activity [138]. Other groups also discovered several small molecules of plant origin exhibiting potent antiviral activity, by pseudotyping *HIV-1* with the *SARS-CoV-2* spike protein harboring the D614G mutation [138].

### 4.5. Analysis of Phenotypic Properties from Virus Mutants

Virus structure, infectivity, immunogenicity, and pathogenicity can be greatly altered by small changes in the amino acid sequence or post-translational modifications of viral proteins, especially for those involved in membrane recognition and fusion, or harboring immunodominant epitopes. Point mutations and recombination in the viral genome can act as drivers of genetic variation and shape their evolution and the emergence of new variants. Many research lines involving pseudotyped viruses have in-depth studied acquired phenotypic properties through genetic polymorphisms.

Early in the COVID-19 pandemic, the spike D614G mutation was shown to increase infectivity, although without affecting neutralization sensitivity of pseudoviruses [139]. *SARS-CoV-2* VLP pseudotypes were employed to demonstrate why the *Delta* variant showed enhanced viral fitness. Findings suggested that mutations within the N protein increased replication in lung epithelial cells by altering viral mRNA packaging and expression [31]. The altered antigenic features and immune evasion properties of the *Omicron* variant have also been extensively studied using *SARS-CoV-2* spike pseudotypes, which have been crucial to inform third-dose vaccination programs. Authors evidenced that the *Omicron* variant spike protein contains mutations that confer evasion from vaccine-elicited neutralizing antibodies, especially for the ChAdOx-1 vaccine, while also shifting tissue tropism away from TMPRSS2-expressing cells, leading to a reduced pathogenesis that explains milder clinical symptoms [51]. Point mutations and altered glycosylation patterns studied with *EBOV Makona* GP pseudotypes were shown to affect entry and increase tropism for human cells, favoring human-to-human transmission, while simultaneously reducing tropism for fruit bat cells. These adaptive mutations could have contributed to the extended duration of the fatal 2013–2016 outbreak, paired with pure epidemiological factors [71]. Similar pseudotyped-based analyses provided further evidence of specific mutations in the *EBOV Makona* GP that may have played a role in its increased virulence and pathogenicity during that outbreak [73].

## 5. Discussion

In the last 5–6 years, public health has witnessed the long-awaited approval of several vaccines and monoclonal antibodies to prevent viral pathogens that cause a high disease burden and mortality, especially in pediatric populations and low-income countries: *RSV*, *Dengue*, *Chikungunya*, *mpox*, *EBOV*, *SARS-CoV-2*, etc. Until this time, prophylactic interventions were either very restricted to specific risk patients and of low efficacy or directly non-existent. For other viral threats such as pandemic *H5N1 avian Influenza*, *Norovirus*, or *Lassa* fever, their vaccine development clinical programs are currently ongoing and nearing licensure. Despite this remarkable progress, virologists in both academia and the pharmaceutical industry involved in drug discovery still face significant challenges, such as prohibitive costs for investments, current technological frontiers, or operational issues, all of which slow down or hamper product commercialization.

Historically, live virus experimentation has limited biological research in virology, given the low accessibility of BSL3/4 settings. The need for high-containment laboratories not only imposes financial and logistical barriers but also slows down the pace of antiviral screening, vaccine development, and serological assessments. In contrast, pseudotyped viruses have emerged as a powerful tool that circumvents many of these limitations, enabling safer and more cost-effective virological studies. Their ability to incorporate key antigenic determinants of highly pathogenic viruses, without requiring full replication competency or isolating the native strain, makes them ideal for evaluating specific neutralizing antibody responses while significantly reducing biosafety concerns and associated costs.

However, pseudotyped systems also involve several limitations, as in many cases they fail to reproduce the behavior of live viruses. One major concern is that most pseudotyped systems focus solely on surface glycoproteins (e.g., the spike protein in coronaviruses), potentially neglecting the contribution of antibodies targeting non-spike viral proteins that may play protective roles. This narrow antigenic representation may lead to an underestimation of the full neutralizing capacity of a serum sample. Moreover, the structural features of pseudotyped viruses often differ from their wild-type counterparts. For instance, the spike density on pseudotyped virions can be lower or display altered spatial organization compared to authentic virions, as some have a preference for specific membrane domains during budding, potentially affecting antibody avidity and neutralization sensitivity [15,19]. These structural discrepancies may misrepresent how effectively antibodies neutralize in a physiological context. Furthermore, pseudotyped viruses do not typically allow for the assessment of immune functions beyond entry inhibition, such as antibody-mediated neutralization of cell-to-cell spread, or post-entry effects like those seen in antibody-dependent intracellular neutralization. They also fail to detect the role of Fc-mediated functions such as antibody-dependent enhancement (ADE) or cytotoxicity. These factors underscore that, while pseudotyped viruses are invaluable tools, especially for their convenient use in BSL-2 settings, their limitations in preclinical or clinical evaluation may be complemented by data from live virus assays or other immunological assessments to ensure a comprehensive understanding of protective immunity.

For most viruses, neutralizing antibody levels provide relevant information to understand vaccine-induced or natural disease protection [4,52], since they can not only inhibit virus infection but also mediate additional immunomodulatory functions such as complement deposition and phagocytic activity [140]. The use of pseudotyped viruses in neutralization assays has streamlined the assessment of these responses, allowing researchers to rapidly characterize antibody potency against emerging variants without the risks and expenses tied to handling live viruses. However, before any serological assay using pseudotypes is widely implemented for diagnostic purposes or vaccine evaluation in clinical trials, it must undergo rigorous validation and standardization to gain acceptance by public health organizations. Non-profit organizations such as the Coalition for Epidemic Preparedness Innovations (CEPI), paired with the WHO, have established international centralized laboratory networks aimed at standardizing immunological assays as pseudotyped neutralization assays, providing testing support to vaccine developers, identifying the immune correlates of protection for vaccines, and facilitating reliable head-to-head comparison. Its pivotal work has boosted support to many vaccine developers worldwide embarked on ambitious vaccine initiatives for priority pathogens (*Marburg-*, *Ebola-*, *Chikungunya-*, *Lassa-*, *Nipah-* virus, etc.), enhancing product licensure and evaluation processes to allow faster roll-out of safe and effective measures to tackle current and future epidemics [141].

As we have reviewed, the wide diversity of viral vectors has allowed researchers to select the most suitable platform depending on the research objective and the structural proteins of the pseudotyped protein. Lentiviral vectors are still more frequently used today when studying neutralization responses, due to their efficient incorporation of viral glycoproteins, stable expression, and reliable infectivity measurements. *MLV* or *VSV* pseudotypes may offer alternative options more advantageous for specific glycoproteins or experimental conditions, showing a higher correlation with live-virus assays, such as the case of *Ebolavirus* GP [98]. Therefore, the choice of vectors should not be universal and depends on factors such as particle stability, tropism, target cell lines, application, and the reliability of existing standardized assays.

Looking to the future, the field of pseudotyped viruses holds immense transformative potential. As our comprehension of the biology of certain viruses deepens and our tools for genetic manipulation become more sophisticated, the scope and success of viral vectors will likely continue to broaden. The strategic use of pseudotyped viruses will remain central to understanding immune protection, viral entry mechanisms, and therapeutic interventions for emerging variants. Future advancements in vector engineering and assay standardization will further refine their applications, ensuring they remain at the forefront of viral immunology, vaccine evaluation, and antiviral drug discovery. We hope that this review provides not only virologists and vaccinologists but also epidemiologists, clinicians, and policymakers with a comprehensive understanding of the current state of research in pseudotyped viruses and especially their role in understanding disease protection against viral pathogens. We intend that this work may be useful for future advancements with clinical impact in infectious diseases.

## Figures and Tables

**Figure 1 microorganisms-13-01785-f001:**
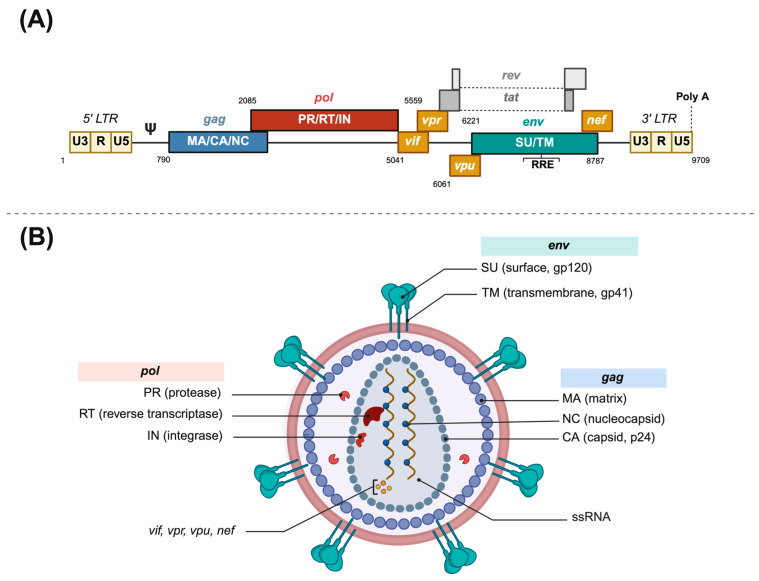
Human immunodeficiency virus structure. (**A**) *HIV-1* provirus genome structure. (**B**) *HIV-1* virion structure. *LTR*: Long Terminal Repeat. *RRE*: rev responsive element. ssRNA: single-stranded RNA. Poly A: polyadenine terminal sequence. Made with Biorender.

**Figure 2 microorganisms-13-01785-f002:**
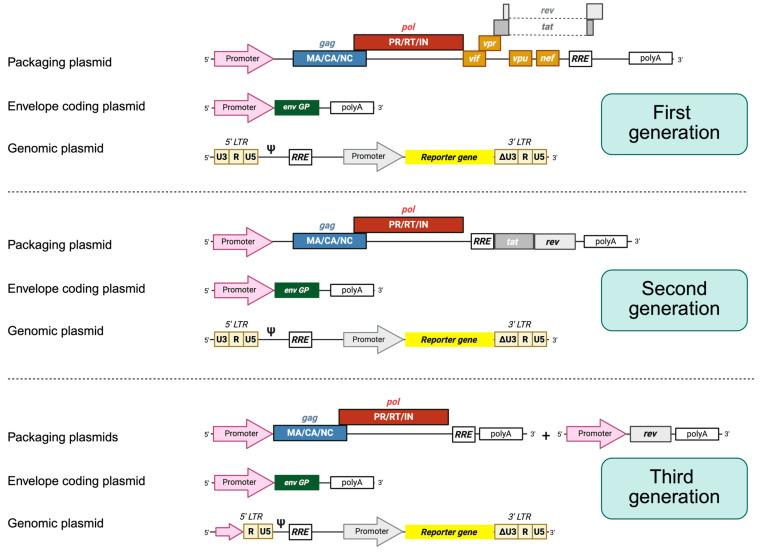
*HIV-1* lentiviral packaging constructs and vectors. In general, three plasmids are employed. First-generation packaging constructs encode *gag*, *pol*, *vif*, *tat*, *rev*, *nef*, and *vpr* proteins under the control of an external promoter, usually *CMV*. The envelope protein (*Env*) is always encoded by another expression plasmid. The plasmid leading to the synthesis of the genomic RNA contains the sequences required in cis for the packaging and reverse transcription of the RNA, also harboring the sequence of a reporter gene under the control of an internal promoter. The expression of the genomic RNA is driven by the 5′ end *LTRs*. In advanced self-inactivating vectors (unshown), the U3 sequence is fully deleted, inactivating the promoter. Second-generation packaging constructs encode only *gag*, *pol*, *tat*, and *rev*. Third-generation packaging constructs are split into two plasmids, where one encodes *gag* and *pol* genes and the other one encodes the *rev* protein.

**Figure 3 microorganisms-13-01785-f003:**
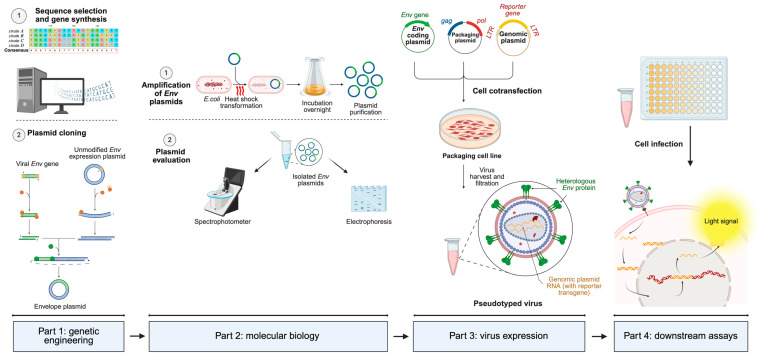
Generic operational workflow of pseudotyped virus generation using a 3-plasmid co-transfection system. The first part involves gene sequence selection and modification (if necessary), extracted from online genomic databases (GISAID, National Center for Biotechnology Information, etc.). The next step consists of cloning the selected gene into an adequate mammalian expression vector. The second part involves amplification of cloned plasmids in bacteria, followed by plasmid isolation, quantification through the spectrophotometer, and evaluation of its integrity and molecular weight through electrophoresis. The third part involves eukaryotic packaging cell transfection with the whole lentiviral system, in order to recover fully infectious pseudotyped viruses harboring the heterologous glycoprotein and the genomic RNA. The fourth and last part involves cell infection with the generated pseudotype. When the pseudotyped virus infects the cell, genomic RNA containing the reporter gene is retrotranscribed, translocated into the nucleus, integrated into the host genome, and translated to produce a measurable light signal. Based on the infective capacity, several different procedures, which vary significantly among each other, can be applied and are not shown for simplification purposes. Made with Biorender.

**Figure 4 microorganisms-13-01785-f004:**
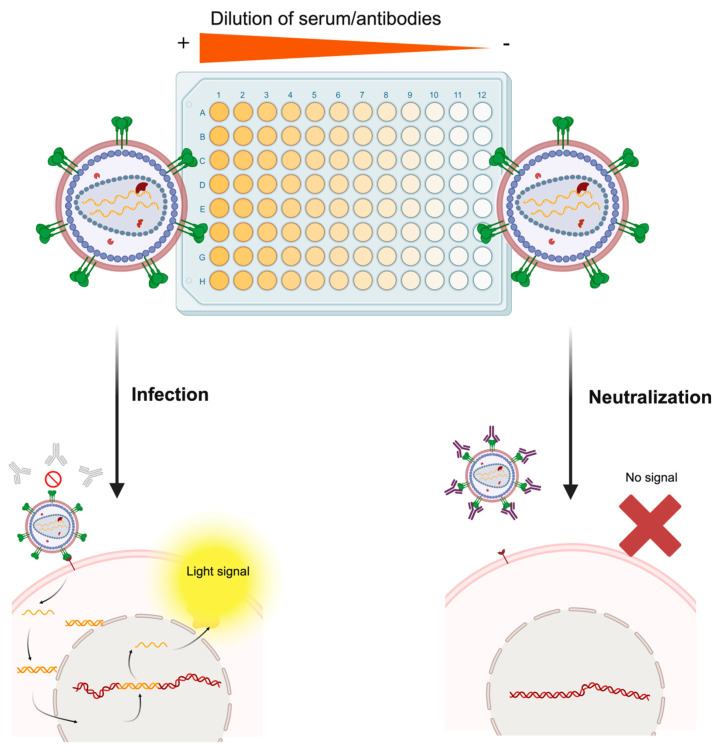
Molecular basis of the pseudotype-based neutralization assay. The serum sample of an individual following vaccination or natural infection is serially diluted in the microtiter plate. Then, the infectious pseudotype suspension is added to each well. In wells containing serum with non-neutralizing antibodies or extremely diluted samples (right part of the figure), the pseudotyped virus will bind the specific host cell receptor of the pseudotyped protein and infect the cell, which will trigger expression of the reporter gene and render a measurable light signal. On the contrary, in wells containing high levels of neutralizing antibodies (left part of the figure), the pseudotyped virus will be neutralized by antibodies, and no infectious process will be established, thus preventing the cell from expressing the reporter gene. Rather than this simplistic dichotomy, this assay generates a continuum of relative light unit (RLU) values, which are plotted as a neutralization curve, each one corresponding to a specific dilution factor. Made with Biorender.

## Data Availability

No new data were created or analyzed in this study. Data sharing is not applicable to this article.

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
