# Peer review of "Exploring the Use of Viral Vectors Pseudotyped with Viral Glycoproteins as Tools to Study Antibody-Mediated Neutralizing Activity"

_microorganisms, 2025, doi:10.3390/microorganisms13081785_

Round 1
Reviewer 1 Report
Comments and Suggestions for Authors
General considerations:
As stated in the title and abstract of this stimulating review, the focus of this work is on the use of pseudotyped viruses as surrogates for highly virulent wild-type viruses in the context of virus neutralization assays, vaccine development, and testing of antivirals that interfere with virus attachment and entry into host cells.
My impression is that the authors were carried away a bit by their justified enthusiasm for these tools and lost focus on their work, meandering in the wide field of recombinant viruses, especially those used in gene therapy, which are already reviewed in specialized articles. The authors noted this pitfall, and I quote: «4.6. Gene Therapy: Although gene therapy exceeds the scope of this review and viral vectors employed for this purpose are not homogenously classified as pseudotyped viruses, it may be worthwhile to briefly outline some significant examples owing to their recent clinical success (7).»
My advice would be to refocus this good work on the original aim, trimming all the parts that are not strictly relevant.
On the other hand, assuming the focus is on virus neutralization, I missed a critical assessment of the use of pseudotyped viruses, which, in many instances, fail to reproduce the behavior of wild-type viruses. For example, the risk of missing neutralizing or protective antibodies targeting non-spike proteins, the concerns about spike density on pseudotyped viruses, which may alter antibody avidity and neutralization sensitivity, missing antibodies that may neutralize a virus in a post-entry phase, missing antibodies that block cell to cell spread, missing antibodies mediating antibody-dependent enhancement.
Detailed criticism:
Page 1: What do you mean by «… that new pandemic may persist over time? »
Comment: Is it the danger of having new pandemics that persist, or do you mean, such as in the case of HIV and SARS-CoV-2, that the pandemic virus will persist?
Page 2: « Vaccine-induced immunity primarily protects through generation of antibodies, because most pathogens reach their target tissues by crossing through the bloodstream in an extracellular state. »
Comment: I am not sure that this sentence is correct. This is undoubtedly true for viruses that undergo a viremic phase; however, what about the majority of respiratory viruses or neurotropic viruses that enter the CNS via axonal transport?
«Broadly speaking, any kind of recombinant virus bearing a foreign gene, regardless of expressing a foreign protein on the virion surface, can be classified as a pseudotyped virus.»
Comment: I am not sure that I can agree with this statement. Most articles concerning pseudotyped viruses focus on the expression of a “foreign” surface protein that interacts with a specific receptor.
«This foreign protein is normally the one that mainly mediates entry into cells and contains the epitopes that trigger the main host immune response, while its functionality remains (ideally) unchanged.»
Comment: It may be better to speak about mediating cell binding and entry.
Page 3: «Lentiviral vectors are expressed by cotransfecting eukaryotic cells are cotransfected with a plasmid coding a truncated form of the genomic RNA, and additional transcomplementation plasmids coding for structural proteins.»
Comment: This sentence should be revised.
Page 6: «Sometimes, the aforementioned vectors have been proven useless when pseudotyping certain envelope proteins from relevant human pathogens, at least with acceptable viral titres.»
Comment: Do you mean: … when pseudotyping the virus with specific envelope proteins…
Page 9: «These findings help explain Omicron's altered infectivity, fusogenicity, and potentially lower severity compared to Delta, while also highlighting the retained efficacy of antiviral drugs like remdesivir and molnupiravir (44).»
Comment: The retained efficacy of antiviral drugs is not related to the activity of the S protein.
Page 12: «Viral diversity can be reproduced by generating pseudotype libraries, which can be useful to evaluate broad-spectrum vaccines or the effects of antibody cross-reactivity on clinical outcome following natural infection, immunotherapy, or vaccination (32,33,91–93).»
Comment: The effects of antibody cross-reactivity on clinical outcome cannot be assessed using pseudotyped viruses in vitro. Maybe to predict the potential protective activity of these antibodies.
Page 13: «Virus-like particles have provided robust vaccine platforms, especially for non enveloped viruses, which for many diseases have been superior in terms of safety and efficacy than classic replicative, inactivated, or subunit vaccines. Thriving VLP uses as immunogens include worldwide implemented vaccines for HPV and HBV, still considered as state-of-the-art and a paradigm of immune durability and vaccine effectiveness in real-world settings (29).»
Comment: I don’t think that VLPs in general can be considered pseudotyped viruses.
Page 15: «The altered antigenic features and immune evasion properties of the Omicron variant have also been extensively studied using SARS-CoV-2 Spike pseudotypes, which has been crucial to inform third-dose vaccination programs and therapeutic regimes with remdesivir and molnupiravir.»
Comment: I certainly agree with the importance of pseudotyped viruses in promoting third-dose vaccination programs, but what about therapeutic regimens with remdesivir and molnupiravir?
Author Response
General comment: As stated in the title and abstract of this stimulating review, the focus of this work is on the
use of pseudotyped viruses as surrogates for highly virulent wild-type viruses in the
context of virus neutralization assays, vaccine development, and testing of antivirals that
interfere with virus attachment and entry into host cells.
My impression is that the authors were carried away a bit by their justified enthusiasm for
these tools and lost focus on their work, meandering in the wide field of recombinant
viruses, especially those used in gene therapy, which are already reviewed in specialized
articles. The authors noted this pitfall, and I quote: «4.6. Gene Therapy: Although gene
therapy exceeds the scope of this review and viral vectors employed for this purpose are
not homogenously classified as pseudotyped viruses, it may be worthwhile to briefly
outline some significant examples owing to their recent clinical success (7).»
My advice would be to refocus this good work on the original aim, trimming all the parts
that are not strictly relevant.
On the other hand, assuming the focus is on virus neutralization, I missed a critical
assessment of the use of pseudotyped viruses, which, in many instances, fail to reproduce
the behavior of wild-type viruses. For example, the risk of missing neutralizing or protective
antibodies targeting non-spike proteins, the concerns about spike density on pseudotyped
viruses, which may alter antibody avidity and neutralization sensitivity, missing antibodies
that may neutralize a virus in a post-entry phase, missing antibodies that block cell to cell
spread, missing antibodies mediating antibody-dependent enhancement.
Response: to refocus the article in the use of pseudotyped viruses for antibody-mediated neutralizing activity and entry inhibitors, the full section of gene/cellular therapy has been removed. Adittionally, those parts mentioning gene therapy in the summary Table 1 or introduction have also been excluded. All other changes are indicated in detail in either blue or red, which each Editor´s comment associated in the margin.
What do you mean by «… that new pandemic may persist over time? »
Comment 1: Is it the danger of having new pandemics that persist, or do you mean, such as
in the case of HIV and SARS-CoV-2, that the pandemic virus will persist?
Response 1: the sentence has been reformulated to "the danger of new pandemics..." as it can be seen in the manuscript.
« Vaccine-induced immunity primarily protects through generation of antibodies,
because most pathogens reach their target tissues by crossing through the bloodstream in
an extracellular state. »
Comment 2: I am not sure that this sentence is correct. This is undoubtedly true for viruses
that undergo a viremic phase; however, what about the majority of respiratory viruses or
neurotropic viruses that enter the CNS via axonal transport?
Response 2: we have now specified that the statement makes reference to viruses that undergo a viremic phase
«Broadly speaking, any kind of recombinant virus bearing a foreign gene, regardless of
expressing a foreign protein on the virion surface, can be classified as a pseudotyped
virus.»
Comment 3: I am not sure that I can agree with this statement. Most articles concerning
pseudotyped viruses focus on the expression of a “foreign” surface protein that interacts
with a specific receptor.
Response 3: this phrase has been removed
«This foreign protein is normally the one that mainly mediates entry into cells and contains
the epitopes that trigger the main host immune response, while its functionality remains
(ideally) unchanged.»
Comment 4: It may be better to speak about mediating cell binding and entry.
Response 4: the part regarding epitopes and host immune response has been removed
Page 3: «Lentiviral vectors are expressed by cotransfecting eukaryotic cells are
cotransfected with a plasmid coding a truncated form of the genomic RNA, and additional
transcomplementation plasmids coding for structural proteins.»
Comment 4: This sentence should be revised.
Response 4: the sentence has been revised and changed to: "Lentiviral vectors are generated by co-transfecting eukaryotic cells with a plasmid coding for a truncated form of the genomic RNA, and additional plasmids coding for structural proteins"
Page 6: «Sometimes, the aforementioned vectors have been proven useless when
pseudotyping certain envelope proteins from relevant human pathogens, at least with
acceptable viral titres.»
Comment 5: Do you mean: … when pseudotyping the virus with specific envelope
proteins…
Response 5: the sentence has been simplified and changed to "Sometimes, the aforementioned vectors have been proven inefficient when pseudotyping the virus with specific envelope proteins from relevant human pathogens"
Page 9: «These findings help explain Omicron's altered infectivity, fusogenicity, and
potentially lower severity compared to Delta, while also highlighting the retained efficacy of
antiviral drugs like remdesivir and molnupiravir (44).»
Comment 7: The retained efficacy of antiviral drugs is not related to the activity of the S
protein.
Response 7: the part regarding the efficacy of antiviral drugs has been removed
Page 12: «Viral diversity can be reproduced by generating pseudotype libraries, which can
be useful to evaluate broad-spectrum vaccines or the effects of antibody cross-reactivity
on clinical outcome following natural infection, immunotherapy, or vaccination
(32,33,91–93).»
Comment: The effects of antibody cross-reactivity on clinical outcome cannot be assessed
using pseudotyped viruses in vitro. Maybe to predict the potential protective activity of
these antibodies.
Response 8: the part of antibody cross-reactivity on clinical outcome has been removed, and the sentence has been modified to: "Viral diversity can be reproduced by generating pseudotype libraries, which can be useful to evaluate broad-spectrum vaccines or to predict the potential protective activity of neutralizing antibodies
Page 13: «Virus-like particles have provided robust vaccine platforms, especially for non
enveloped viruses, which for many diseases have been superior in terms of safety and efficacy than classic replicative, inactivated, or subunit vaccines. Thriving VLP uses as immunogens include worldwide implemented vaccines for HPV and HBV, still considered as state-of-the-art and a paradigm of immune durability and vaccine effectiveness in real-world settings (29).»
Comment: I don’t think that VLPs in general can be considered pseudotyped viruses.
Response 9: this part and other regarding VLPs have been removed
Page 15: «The altered antigenic features and immune evasion properties of the Omicron
variant have also been extensively studied using SARS-CoV-2 Spike pseudotypes, which
has been crucial to inform third-dose vaccination programs and therapeutic regimes with
remdesivir and molnupiravir.»
Comment: I certainly agree with the importance of pseudotyped viruses in promoting third-
dose vaccination programs, but what about therapeutic regimens with remdesivir and
molnupiravir?
Response 10: the part regarding therapeutic regimes with remdesivir and molnupiravir has been fully eliminated
Reviewer 2 Report
Comments and Suggestions for Authors
Dear Authors,
The article “Exploring the Use of Viral Vectors Pseudotyped with Viral Glycoproteins as Tools to Study Antibody-Mediated Neutralizing Activity” introduces the review of pseudotyped viruses, candidate vaccine viruses and antivirals in both preclinical models and clinical trials. The review was done at a high level.
The are some omissions:
- In the introduction section it is desirable to include the advantages of pseudotyped viruses compared to authentic viruses
- Coxsackievirus, Epstein-Barr virus, Hepatitis B and C are not discussed
- It is desirable to discuss recent clinical trials of pediatric patients with inborn errors of immunity and/or post allogeneic hematopoietic stem cell transplant with refractory viral infections using partially-HLA matched VSTs targeting cytomegalovirus, Epstein-Barr virus, or adenovirus.
For example, the ref.: Keller, M.D., Hanley, P.J., Chi, YY. et al. Antiviral cellular therapy for enhancing T-cell reconstitution before or after hematopoietic stem cell transplantation (ACES): a two-arm, open label phase II interventional trial of pediatric patients with risk factor assessment. Nat Commun 15, 3258 (2024). https://doi.org/10.1038/s41467-024-47057-2
- The figures are not referenced in the text. Only Fig 2 is mentioned in the text.
- Please, discuss the development of 1-3d generations of HIV-1 lentiviral packaging constructs and vectors, what are the benefits of the 3d generation.
- It is desirable to discuss the limitations and side effects of gene antiviral therapy

Author Response
All changes can be seen marked in red/blue in the updated version of the manuscript. All of the Editor´s comments have also been adressed and indicated there.
Editor 2 comments:
The are some omissions:
1. In the introduction section it is desirable to include the advantages of pseudotyped viruses
compared to authentic viruses
Response 1: a brief paragraph introducing the main features and why pseudotyped viruses are preferable especially for their biosafety has been included, in the introduction section.
2. Coxsackievirus, Epstein-Barr virus, Hepatitis B and C are not discussed
Response 2: Coxsackievirus, HBV, and HCV have all now been included in the text and in the summary Table 1. Regarding E-B virus, since most of the literature of pseudotyped viruses was related with gene therapy and this section has been fully excluded following the Editor´s 1 advice, no mentions to this virus have been made in the text.
3. It is desirable to discuss recent clinical trials of pediatric patients with inborn errors of immunity
and/or post allogeneic hematopoietic stem cell transplant with refractory viral infections using
partially-HLA matched VSTs targeting cytomegalovirus, Epstein-Barr virus, or adenovirus.
For example, the ref.: Keller, M.D., Hanley, P.J., Chi, YY. et al. Antiviral cellular therapy for
enhancing T-cell reconstitution before or after hematopoietic stem cell transplantation (ACES): a
two-arm, open label phase II interventional trial of pediatric patients with risk factor
assessment. Nat Commun 15, 3258 (2024). https://doi.org/10.1038/s41467-024-47057-2
Response 3: all of the gene/cellular therapy section has been excluded following editor´s 1 advice, therefore no additional references or paragraphs have been included here.
4. The figures are not referenced in the text. Only Fig 2 is mentioned in the text.
Response 4: all figures have now been mentioned at least once in the main text
5. Please, discuss the development of 1-3d generations of HIV-1 lentiviral packaging constructs and
vectors, what are the benefits of the 3d generation.
Response 5: a new section discussing the history and development of lentiviral vectors has been now included in the section 3.1. Two new references have also been included for this purpose. The exact text is as follows: "Each of the three generations or HIV-1 derived lentiviral vectors aimed to address specific biosafety and functional concerns inherent to working with a human retrovirus. The first-generation packaging systems maintained most of the HIV-1 genome across only two or three plasmids and included several accessory genes such as vif, vpr, vpu, and nef, raising significant biosafety concerns due to the potential for recombination and genera- tion of replication-competent lentiviruses (13). For second-generation vectors, these non- essential accessory genes were removed, and the tat gene was retained, with transcription still relying on the HIV-1 LTR promoter. This version improved biosafety while maintain- ing vector efficiency. The major leap in safety and modularity came with third-generation vectors, which introduced several innovations: tat dependence was eliminated by replac- ing the HIV-1 LTR promoter with a constitutively active promoter (such as CMV or RSV) the rev gene was segregated onto a separate plasmid, and the structural and enzymatic genes (gag, pol) were further separated from the envelope gene, resulting in a four-plas- mid system. Additionally, self-inactivating (SIN) LTRs were introduced, preventing reac- tivation of the integrated provirus and reducing the risk of insertional mutagenesis (13).
Further safety and performance features were incorporated over time to third gener- ation systems, such as the Rev Response Element (RRE) for efficient nuclear export, central polypurine tract (cPPT) for improved nuclear import, and Woodchuck Pos]ranscriptional Regulatory Element (WPRE) to enhance transgene expression. These advancements not only greatly minimized the possibility of replication-competent lentiviruses formation, but also improved transduction efficiency in non-dividing cells (a current hallmark of len- tiviral systems). As a result, third-generation lentiviruses are now considered the safest and most versatile standard for a wide range of molecular biology, immunology, and cell engineering applications, offering a strong balance between high-level expression and regulatory compliance (14)."
6. It is desirable to discuss the limitations and side effects of gene antiviral therapy
Response 6: all of the gene/cellular therapy section has been excluded following editor´s 1 advice, therefore no additional references or paragraphs have been included here.
Round 2
Reviewer 1 Report
Comments and Suggestions for Authors
The authors have responded convincingly to my questions, and from my point of view, this article can be published in its current form.
Reviewer 2 Report
Comments and Suggestions for Authors
Dear Authors,
The article has been improved and can be accepted for publication